# Empirical Study of Off-Policy Policy Evaluation for Reinforcement Learning

**Cameron Voloshin**
Caltech

**Hoang M. Le**
Argo AI

**Nan Jiang**
UIUC

**Yisong Yue**
Caltech

## Abstract

We offer an experimental benchmark and empirical study for off-policy policy evaluation (OPE) in reinforcement learning, which is a key problem in many safety critical applications. Given the increasing interest in deploying learning-based methods, there has been a flurry of recent proposals for OPE method, leading to a need for standardized empirical analyses. Our work takes a strong focus on diversity of experimental design to enable stress testing of OPE methods. We provide a comprehensive benchmarking suite to study the interplay of different attributes on method performance. We also distill the results into a summarized set of guidelines for OPE in practice. Our software package, the Caltech OPE Benchmarking Suite (COBS), is open-sourced and we invite interested researchers to further contribute to the benchmark.

## 1 Introduction

Reliably leveraging logged data for decision making is an important milestone for realizing the full potential of reinforcement learning. A key component is the problem of off-policy policy evaluation (OPE), which aims to estimate the value of a target policy using only pre-collected historical (logging) data generated by other policies. Given its importance, the research community actively advances OPE techniques, both for the bandit [15, 3, 49, 56, 32, 36] and reinforcement learning settings [26, 15, 16, 33, 58, 54, 40, 59, 10]. These new developments reflect practical interests in deploying reinforcement learning to safety-critical situations [31, 57, 3, 1], and the increasing importance of off-policy learning and counterfactual reasoning [12, 52, 37, 30, 34, 41]. OPE is also similar to the dynamic treatment regime problem in the causal inference literature [39].

In this paper, we present the **Caltech OPE Benchmarking Suite (COBS)**, which benchmarks OPE techniques via experimental designs that give thorough considerations to factors that influence performance. The reality of method performance, as we will discuss, is nuanced and comparison among different estimators is tricky without pushing the experimental conditions along various dimensions. Our philosophy and contributions can be summarized as follows:

- We establish a benchmarking methodology that considers key factors that influence OPE performance, and design a set of domains and experiments to systematically expose these factors. The proposed experimental domains are complementary to continuous control domains from recent offline RL benchmarks [18, 17]. We differ from these recent benchmarks in two important ways:
  1. COBS allows researchers fine-grained control over experimental design, other than just access to a pre-collected dataset. The offline data can be generated "on-the-fly" based on experimental criteria, e.g., the divergence between behavior and target policies.
  2. We offer significant diversity in experimental domains, covering a wide range of dimensionality and stochasticity. Together, the goal of this greater level of access is to enable a deeper look at when and why certain methods work well.
- As a case study, we select a representative set of established OPE baseline methods, and test them systematically. We further show how to distill the empirical findings into key insights to guide practitioners and inform researchers on directions for future exploration.

35th Conference on Neural Information Processing Systems (NeurIPS 2021) Track on Datasets and Benchmarks.

- COBS is an extensive software package that can interface with new environments and methods to run new OPE experiments at scale.[1] Given the fast-changing nature of this active area of research, our package is designed to accommodate the rapidly growing body of OPE estimators. COBS is already actively used by multiple research groups to benchmark new algorithms.

**Prior Work.** Empirical benchmarks have long contributed to the scientific understanding, advancement, and validation of machine learning techniques [8, 6, 7, 46, 14, 13]. Recently, many have called for careful examination of empirical findings of contemporary deep learning and deep reinforcement learning efforts [23, 35]. As OPE is central to real-world applications of reinforcement learning, proper benchmarking is critical to ensure in-depth understanding and accelerate progress. While many recent methods are built on sound mathematical principles, a notable gap in the current literature is a standard for benchmarking empirical studies, with perhaps a notable exception from the recent DOPE [18] and D4RL benchmarks [17].

Compared to prior complementary work on OPE evaluation for reinforcement learning [17, 18], our benchmark offers two main advantages. First, we focus on maximizing reproducibility and nuanced experimental control with minimal effort, covering data generation and fine-grained control over factors such as relative "distance" between the offline data distribution and the distribution induced by evaluation policies. Second, we study a diverse set of environments, spanning range of desiderata such as stochastic-vs-deterministic and different representations for the same underlying environment. Together, these attributes enable our benchmarking suite to conduct systematic analyses of the method performance under different scenarios, and provide a holistic summary of the challenges one may encounter in different scenarios.

**Background & Notation.** As per RL standard, we represent the environment by $\langle X, A, P, R, \gamma \rangle$. $X$ is the state space (or observation space in the non-Markov case). OPE is typically considered in the episodic RL setting. A behavior policy $\pi_b$ generates a historical data set, $D = \{\tau^i\}_{i=1}^N$, of $N$ trajectories (or episodes), where $i$ indexes over trajectories, and $\tau^i = (x_0^i, a_0^i, r_0^i, \ldots, x_{T-1}^i, a_{T-1}^i, r_{T-1}^i)$. The episode length $T$ is assumed to be fixed for notational convenience. Given a desired evaluation policy $\pi_e$, the OPE problem is to estimate the value $V(\pi_e)$, defined as: $V(\pi_e) = \mathbb{E}_{x \sim d_0} \left[ \sum_{t=0}^{T-1} \gamma^t r_t | x_0 = x \right]$, with $a_t \sim \pi_e(\cdot | x_t)$, $x_{t+1} \sim P(\cdot | x_t, a_t)$, $r_t \sim R(x_t, a_t)$, and $d_0 \subseteq X$ is the initial state distribution.

## 2  Benchmarking Design & Methodology

### 2.1  Design Philosophy

The design philosophy of the Caltech OPE Benchmarking Suite (COBS) starts with the most prominent decision factors that can make OPE difficult. These factors come from both the existing literature and our own experimental study, which we will further discuss. We then seek to design experimental conditions that cover a diverse range of these factors. As a sub-problem within the broader reinforcement learning problem class, OPE experiments in existing literature gravitate towards commonly used RL domains. Unsurprisingly, the most common experiments belong to the Mujoco group of deterministic continuous control tasks [53], or discrete domains that operate via OpenAI Gym interface [4]. For OPE, high-dimensional domains such as Atari [2] appear less often, but are also natural candidates for OPE testing. We selectively pick from these domains as well as design new domains, with the goal of establishing refined control over the decision factors.

**Design Factors.** We consider several domain characteristics that are often major factors in performance of OPE methods:

1. *Horizon length*. Long horizons can lead to catastrophic failure in some OPE methods due to an exponential blow-up in some of their components [32, 26, 33].
2. *Reward sparsity*. Sparse rewards represent a difficult credit assignment problem in RL. This factor is often not emphasized in OPE, and arguably goes hand-in-hand with horizon length.[2]
3. *Environment stochasticity*. Popular RL domains such as Mujoco and Atari are mostly deterministic. This is a fundamental limitation in many existing empirical studies since many theoretical

---

[1] https://github.com/clvoloshin/COBS
[2] Considered in isolation, long horizons may not be an issue if the reward signal is dense.

challenges to RL only surface in a stochastic setting. A concrete example is the famous double sampling problem [11], which is not applicable in many contemporary RL benchmarks.

4. *Unknown behavior policy*. This is related to the source of the collected data. The data may come from one or a more policies which may not be known. For example, existing dataset benchmarks, such as D4RL [17] can be considered to come from an unknown behavior policy. Some methods will require behavior policy estimation, thus introducing some bias.

5. *Policy and distribution mismatch*. The relative difference between the evaluation and behavior policy can play a critical role in the performance of many OPE methods. This difference induces a distribution mismatch between the dataset $D$ and the dataset that would have been produced had we run the evaluation policy. Performing out-of-distribution estimation is a key challenge for robust OPE. We focus on providing a systematic way to stress test OPE methods under this mismatch, which we accomplish by offering a control knob for flexible data generation to induce various degrees of mismatch.

6. *Model misspecification*. Model misspecification refers to the insufficient representation power of the function class used to approximate different objects of interest, whether the transition dynamics, value functions, or state distribution density ratio. In realistic applications, it is reasonable to expect at least some degree of misspecification. We study the effect of misspecification via two controlled scenarios: (i) we start with designing simple domains to test OPE methods under tabular representation and (ii) we test the same OPE methods and same tabular data generation process, but the input representation for OPE methods is now modified to expose the impact of choosing a different function class for representation.

## 2.2 Domains

Ultimately, many of the aforementioned factors are intertwined and their usefulness in evaluating OPE performance cannot be considered in isolation. However, they serve as a valuable guide in our selection of benchmark environments. To that end, our benchmark suite includes eight environments. We use two standard RL benchmarks from OpenAI [5]. As many standard RL benchmarks are fixed and deterministic, we design six additional environments that allow control over different design factors. Figure 1 depicts one such design factor: the representation complexity.

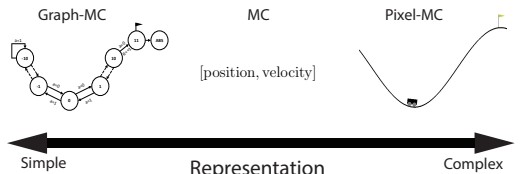

Figure 1: Depicting one of the dimensions which COBS provides control. For the Mountain Car environment, we can select either a tabular, standard coordinate-based, or pixel-based representation of the state while holding other factors fixed.

**Graph:** A flexible discrete environment that can vary in horizon, stochasticity, and sparsity.

**Graph-POMDP:** An extension of Graph to a POMDP setting, where selected information is omitted from the observations that form the behavior data. This enables controlled study of the effect of insufficient representation power relative to other settings in the Graph domain above.

**Gridworld (GW):** A gridworld design that offers larger state and action space than the Graph domains, longer horizon, and similarly flexible design choices for other environmental factors. Using some version of gridworld is standard across many RL experiments. Gridworld enables simple integration of various designs, and fast data collection.

**Pixel-Gridworld (Pix-GW):** A scaled-up domain from Gridworld which enables pixel-based representation of the state space. While such usage is not standard in existing literature, this design offers compelling advantage over many existing standard RL benchmarks. First, this domain enables simple controlled experiments to understand the impact of high-dimensional representation on OPE performance, where the ground truth of various quantities to be estimated is readily obtainable thanks to the access to underlying simpler grid representation. Second, this domain effectively simulates high-dimensional experiments with easily tuned experimental conditions, e.g., degree of stochasticity. This design freedom is not available with many currently standard RL benchmarks.

**Mountain Car (MC):** A standard control domain, which is known to have challenging credit assignment due to sparsity of the reward. Our benchmark for this standard domains allows for

function approximation to vary between a linear model and feed-forward neural network, in order to highlight the effects of model misspecification.

**Pixel Mountain Car (Pix-MC)**: A modified version of Mountain Car where the state input is pixel-based, testing the methods' ability to work in high dimensional settings.

**Tabular Mountain Car (Graph-MC)** A simplified version of Mountain Car to a graph, allowing us to complete the test for model misspecification by considering the tabular case.

**Atari (Enduro)** A pixel-based Atari domain. Note that all Atari environments are deterministic and high-dimensional. Instead of choosing many different Atari domains to study, we instead opt to select Enduro as the representative Atari environment, due to its sparsity of reward (and commonly regarded as a highly challenging task). All Atari environments share similar interaction protocol, and can be seamlessly integrated into COBS, if desired.

All together, our benchmark consists of 8 environments with characteristics summarized in Table 1. Complete descriptions can be found in Appendix F. All environments have finite action spaces.

Table 1: Environment characteristics

| Environment | Graph | Graph-MC | MC | Pix-MC | Enduro | G-POMDP | GW | Pix-GW |
|---|---|---|---|---|---|---|---|---|
| Is MDP? | yes | yes | yes | yes | yes | no | yes | yes |
| State desc. | pos. | pos. | [pos, vel] | pixels | pixels | pos. | pos. | pixels |
| $T$ | 4 or 16 | 250 | 250 | 250 | 1000 | 2 or 8 | 25 | 25 |
| Stoch Env? | variable | no | no | no | no | no | no | variable |
| Stoch Rew? | variable | no | no | no | no | no | no | no |
| Sparse Rew? | variable | terminal | terminal | terminal | dense | terminal | dense | dense |
| $\hat{Q}$ Class | tabular | tabular | linear/NN | NN | NN | tabular | tabular | NN |
| Initial state | 0 | 0 | variable | variable | gray img | 0 | variable | variable |
| Absorb. state | 2T | 22 | [.5,0] | img([.5,0]) | zero img | 2T | 64 | zero img |
| Frame height | 1 | 1 | 2 | 2 | 4 | 1 | 1 | 1 |
| Frame skip | 1 | 1 | 5 | 5 | 1 | 1 | 1 | 1 |

## 2.3 Experiment Protocol

**Selection of Policies**. We use two classes of policies. The first is state-independent with some probability of taking any available action. For example, in the Graph environment with two actions, $\pi(a = 0) = p, \pi(a = 1) = 1 - p$ where $p$ is a parameter we can control. The second is a state-dependent $\epsilon-$Greedy policy. We train a policy $Q^*$ (using value iteration or DDQN [22]) and then vary the deviation away from the policy. Hence $\epsilon - Greedy(Q^*)$ implies we follow a mixed policy $\pi = \arg\max_a Q^*(x, a)$ with probability $1 - \epsilon$ and uniform with probability $\epsilon$. Here $\epsilon$ is a parameter we can control.

Most OPE methods explicitly require absolute continuity among the policies ($\pi_b > 0$ whenever $\pi_e > 0$). Thus, all policies will remain stochastic with this property maintained.

**Data Generation & Metrics.** Each experiment depends on specifying an environment and its properties, behavior policy $\pi_b$, evaluation policy $\pi_e$, and number of trajectories $N$ from rolling-out $\pi_b$ for historical data. The true on-policy value $V(\pi_e)$ is the Monte-Carlo estimate via $10,000$ rollouts of $\pi_e$. We repeat each experiment $m = 10$ times with different random seeds. We judge the quality of a method via two metrics:

- Relative mean squared error (*Relative MSE*): $\frac{1}{m} \sum_{i=1}^{m} \frac{(\widehat{V}(\pi_e)_i - \frac{1}{m} \sum_{j=1}^{m} V(\pi_e)_j)^2}{(\frac{1}{m} \sum_{j=1}^{m} V(\pi_e)_j)^2}$, which allows a fair comparison across different conditions.[3]

- *Near-top Frequency*: For each experimental condition, we include the number of times each method is within $10\%$ of the best performing one to facilitate aggregate comparison across domains.

**Implementation & Hyperparameters.** COBS allows running experiments at scale and easy integration with new domains and techniques for future research. The package consists of many domains and reference implementations of OPE methods.

---

[3]The metric used in prior OPE work is typically mean squared error: MSE$= \frac{1}{m} \sum_{i=1}^{m} (\widehat{V}(\pi_e)_i - V(\pi_e)_i)^2$.

Hyperparameters are selected based on publication, code release or author consultation. We maintain a consistent set of hyperparameters for each estimator and each environment across experimental conditions (see hyperparameter choice in appendix Table 12).[4]

## 2.4 Baselines

OPE methods were historically categorized into importance sampling methods, direct methods, or doubly robust methods. This demarcation was first introduced for contextual bandits [15], and later extended to the RL setting [26]. Some recent methods have blurred the boundary of these categories. Examples include Retrace($\lambda$) [37] that uses a product of importance weights of multiple time steps for off-policy $Q$ correction, and MAGIC [51] that switches between importance weighting and direct methods. In this benchmark, we propose to group OPE into three similar classes of methods, but with expanded definition for each category: Inverse Propensity Scoring, Direct Methods, and Hybrid Methods. For the current benchmark, we select representative established baselines from each category. Appendix E contains a full description of all methods under consideration.

**Inverse Propensity Scoring (IPS)** We consider the main four variants: Importance Sampling (IS), Per-Decision Importance Sampling (PDIS), Weighted Importance Sampling (WIS) and Per-Decision WIS (PDWIS). IPS has a rich history in statistics [44, 20, 24], with successful crossover to RL [45]. The key idea is to reweight the rewards in the historical data by the importance sampling ratio between $\pi_e$ and $\pi_b$, i.e., how likely a reward is under $\pi_e$ versus $\pi_b$.

**Direct Methods (DM)** While some direct methods make use of importance weight adjustments, a key distinction of direct methods is the focus on regression-based techniques to (more) directly estimate the value functions of the evaluation policy ($Q^{\pi_e}$ or $V^{\pi_e}$). This is an area of very active research with rapidly growing literature. We consider 8 different direct approaches, taken from the following respective families of direct estimators:

*Model-based estimators* Perhaps the most commonly used DM is *Model-based* (also called approximate model, denoted AM), where the transition dynamics, reward function and termination condition are directly estimated from historical data [26, 43]. The resulting learned MDP is then used to compute the value of $\pi_e$, e.g., by Monte-Carlo policy evaluation. There are also some recent variants of the model-based estimator, e.g., [60].

*Value-based estimators* *Fitted Q Evaluation (FQE)* is a model-free counterpart to AM, and is functionally a policy evaluation counterpart to batch Q learning [30]. $\mathbf{Q}^\pi(\lambda)$ *& Retrace($\lambda$) & Tree-Backup($\lambda$)* Several model-free methods originated from off-policy learning settings, but are also natural for OPE. $Q^\pi(\lambda)$ [21] can be viewed as a generalization of FQE that looks to the horizon limit to incorporate the long-term value into the backup step. Retrace($\lambda$) [37] and Tree-Backup($\lambda$) [45] also use full trajectories, but additionally incorporate varying levels of clipped importance weights adjustment. The $\lambda$-dependent term mitigates instability in the backup step, and is selected based on experimental findings of [37].

*Regression-based estimators* *Direct Q Regression (Q-Reg) & More Robust Doubly-Robust (MRDR)* [16] propose two direct methods that make use of cumulative importance weights in deriving the regression estimate for $Q^{\pi_e}$, solved through a quadratic program. MRDR changes the objective of the regression to that of directly minimizing the variance of the Doubly-Robust estimator.

*Minimax-style estimators* [33] recently proposed a method for the infinite horizon setting - we refer to this estimator as IH. While IH can be viewed as a Rao-Blackwellization of the IS estimator, we include it in the DM category because it solves the Bellman equation for state distributions and requires function approximation, which are more characteristic of DM. IH shifts the focus from importance sampling over action sequences to importance ratio between *state density distributions* induced by $\pi_b$ and $\pi_e$. Starting with IH, this style of minimax estimator has recently attracted significant attention in OPE literature, including state-action extension of IH [54, 25] and DICE family of estimators [40, 61, 59, 10]. For our benchmarking purpose, we choose IH as the representative of this family.

**Hybrid Methods (HM)** Hybrid methods subsume doubly robust-like approaches, which combine aspects of both IPS and DM. Standard doubly robust OPE (denoted DR) [26] is an unbiased estimator that leverages DM to decrease the variance of the unbiased estimates produced by importance sampling techniques: Other HM include Weighted Doubly-Robust (WDR) and MAGIC. WDR

---

[4]In practice, hyperparameter tuning is not practical for OPE due to a lack of validation signal.

replaces the importance weights with self-normalized importance weights (similar to WIS). MAGIC introduces adaptive switching between DR and DM; in particular, one can imagine using DR to estimate the value for part of a trajectory and then using DM for the remainder. Using this idea, MAGIC [51] finds an optimal linear combination among a set that varies the switch point between WDR and DM. Note that any DM that returns $\widehat{Q}^{\pi_e}(x, a; \theta)$ yields a set of corresponding DR, WDR, and MAGIC estimators. As a result, we consider 21 hybrid approaches in our experiments.

## 3 Empirical Evaluation

We evaluate 33 different OPE methods by running thousands of experiments across the 8 domains. Due to limited space, we show only the results from selected environmental conditions in the next section. The full detailed results, with highlighted best method in each class, are available in the appendix. The goal of the evaluation is to demonstrate the flexibility of the benchmark suite to systematically test the different factors of influence. We synthesize the results, and then present further considerations and directions for research in Section 4.

### 3.1 What is the best method?

The first important takeaway is that *there is no clear-cut winner*: no single method or method class is consistently the best performer, as multiple environmental factors can influence the accuracy of each estimator. With that caveat in mind, based on the aggregate top performance metrics, we can recommend from our selected methods the following for each method class (See Figure 3 right, appendix Table 15, and appendix Table 3).

**Inverse propensity scoring (IPS).** In practice, weighted importance sampling, which is biased, tends to be more accurate and data-efficient than unbiased basic importance sampling methods. Among the four IPS-based estimators, *PDWIS tends to perform best* (Figure 3 left).

**Direct methods (DM).** Generally, FQE, $Q^\pi(\lambda)$*, and IH tend to perform the best among DM* (appendix Table 3). FQE tends to be more data efficient and is the best method when data is limited (Figure 5). $Q^\pi(\lambda)$ generalizes FQE to multi-step backup, and works particularly well with more data, but is computationally expensive in complex domains. IH is highly competitive in long horizons and with high policy mismatch in a tabular setting (appendix Tables 7, 8). In pixel-based domains, however, choosing a good kernel function for IH is not straightforward, and IH can underperform other DM (appendix Table 11). We provide a numerical comparison among direct methods for tabular (appendix Figure 17) and complex settings (Figure 3 center).

**Hybrid methods (HM).** With the exception of IH, each DM corresponds to three HM: standard doubly robust (DR), weighted doubly robust (WDR), and MAGIC. *For each DM, its WDR version often outperforms its DR version*. MAGIC can often outperform WDR and DR. However, MAGIC comes with additional hyperparameters, as one needs to specify the set of partial trajectory length to be considered. Unsurprisingly, their performance highly depends on the underlying DM. In our experiments, FQE and $Q^\pi(\lambda)$ are typically the most reliable: *MAGIC with FQE or MAGIC with $Q^\pi(\lambda)$ tend to be among the best hybrid methods* (see appendix Figures 23 - 27).

### 3.2 A recipe for method selection

Figure 2 summarizes our general guideline for navigating key factors that affect the accuracy of different estimators. To guide the readers through the process, we now dive further into our experimental design to test various factors, and discuss the resulting insights.

**Do we potentially have representation mismatch?** Representation mismatch comes from two sources: model misspecification and poor generalization. Model misspecification refers to the insufficient representation power of the function class used to approximate either the transition dynamics (AM), value function (other DM), or state distribution density ratio (in IH).

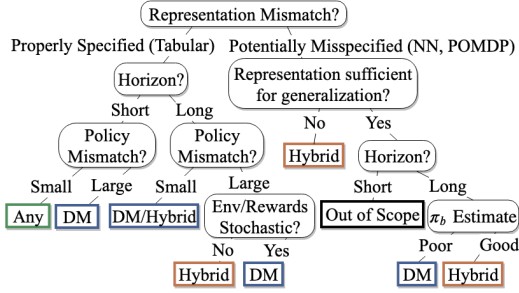

Figure 2: *General Guideline Decision Tree.*

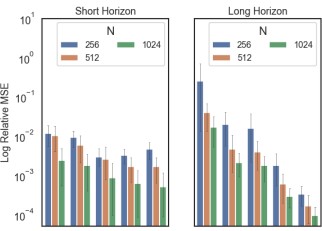
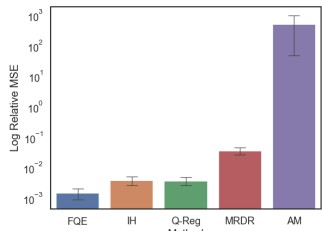

| Method | Near-top Freq. |
|---|---|
| MAGIC FQE | 30.0% |
| DM FQE | 23.7% |
| IH | 19.0% |
| WDR FQE | 17.8% |
| MAGIC $Q^\pi(\lambda)$ | 17.3% |

Figure 3: Left: *(Graph domain) Comparing IPS (and IH) under short and long horizon. Mild policy mismatch setting. PDWIS is often best among IPS. But IH outperforms in long horizon.* Center: *(Pixel-MC) Comparing direct methods in high-dimensional, long horizon setting. Relatively large policy mismatch. FQE and IH tend to outperform. AM is significantly worse in complex domains. Retrace(λ), Q(λ) and Tree-Backup(λ) are very computationally expensive and thus excluded.* Right: *(Top Methods) The top 5 methods which perform the best across all conditions and domains.*

Having a tabular representation controls for representation mismatch by ensuring adequate function class capacity, as well as zero inherent Bellman error (left branch, Fig 2). In such cases, we may still suffer from poor generalization without sufficient data coverage, which depends on other factors in the domain settings.

The effect of representation mismatch (right branch, Fig 2) can be understood via two scenarios:

- *Misspecified and poor generalization:* We expose the impact of this severe mismatch scenario via the Graph POMDP construction, where selected information are omitted from an otherwise equivalent Graph MDP. Here, HM substantially outperforms DM (Figure 4 right versus left).
- *Misspecified but good generalization:* Function classes such as neural networks have powerful generalization ability, but may introduce bias and inherent Bellman error[5] [38, 9] (see linear vs. neural networks comparison for Mountain Car in appendix Fig 14). Still, powerful function approximation makes (biased) DM very competitive with HM, especially under limited data and in complex domains (see pixel-Gridworld in appendix Fig 28-30). However, function approximation bias may cause serious problems for high dimensional and long horizon settings. In the extreme case of Enduro (very long horizon and sparse rewards), all DM fail to convincingly outperform a naïve average of behavior data (appendix Fig 13).

**Short horizon vs. Long horizon?** It is well-known that IPS methods are sensitive to trajectory length [32]. Long horizon leads to an exponential blow-up of the importance sampling term, and is exacerbated by significant mismatch between $\pi_b$ and $\pi_e$. This issue is inevitable for any unbiased estimator [26] (a.k.a., the curse of horizon [33]). Similar to IPS, DM relying on importance weights also suffer in long horizons (appendix Fig 17), though to a lesser degree. IH aims to bypass the effect of cumulative weighting in long horizons, and indeed performs substantially better than IPS methods in very long horizon domains (Fig 3 left).

A frequently ignored aspect in previous OPE work is a proper distinction between fixed, finite horizon tasks (IPS focus), infinite horizon tasks (IH focus), and indefinite horizon tasks, where the trajectory length is finite but varies depending on the policy. Many applications should properly belong to the indefinite horizon category.[6] Applying HM in this setting requires proper padding of the rewards (without altering the value function in the infinite horizon limit) as DR correction typically assumes fixed length trajectories.

**How different are behavior and target policies?** Similar to IPS, the performance of DM is negatively correlated with the degree of policy mismatch. Figure 5 shows the interplay of increasing policy mismatch and historical data size, on the top DM in the deterministic gridworld. We use $(\sup_{a \in A, x \in X} \frac{\pi_e(a|x)}{\pi_b(a|x)})^T$ as an environment-independent metric of mismatch between the two policies. The performance of the top DM (FQE, $Q^\pi(\lambda)$, IH) tend to hold up better than IPS methods when the policy gap increases (appendix Figure 19). FQE and IH are best in the small data regime, and $Q^\pi(\lambda)$ performs better as data size increases (Figure 5). Increased policy mismatch weakens the DM that use importance weights (Q-Reg, MRDR, Retrace(λ) and Tree-Backup(λ)).

---

[5]Inherent Bellman error is defined as $\sup_{g \in F} \inf_{f \in F} ||f - \mathbb{T}^\pi g||_{d_\pi}$, where F is function class chosen for approximation, and $d_\pi$ is state distribution induced by evaluation policy $\pi$.

[6]Applying IH in the indefinite horizon case requires setting up an absorbing state that loops over itself with zero terminal reward.

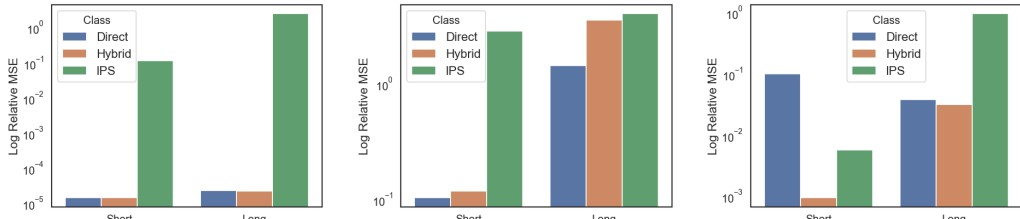

**Figure 4:** *Comparing IPS versus Direct methods versus Hybrid methods under short and long horizon, large policy mismatch and large data.* Left: *(Graph domain) Deterministic environment.* Center: *(Graph domain) Stochastic environment and rewards.* Right: *(Graph-POMDP) Model misspecification (POMDP). Minimum error per class is shown.*

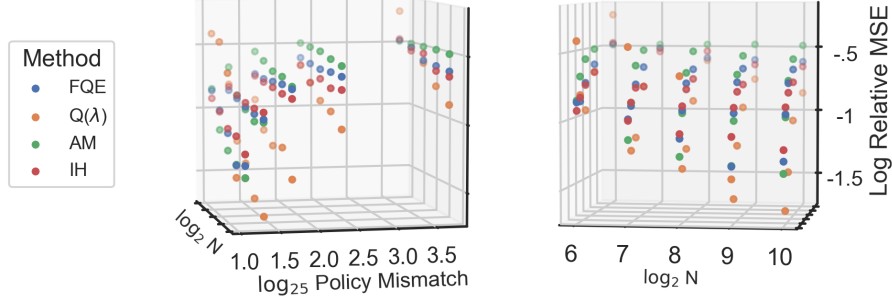

**Figure 5:** *(Gridworld domain) Errors are directly correlated with policy mismatch but inversely correlated with data size. We pick the best direct methods for illustration. The two plots represent the same figure from two different vantage points.*

**Do we have a good estimate of the behavior policy?** Often the behavior policy may not be known exactly and requires estimation, which can introduce bias and cause HM to underperform DM, especially in low data regime (e.g., pixel gridworld appendix Figure 28, 30). Similar phenomenon was observed in the statistics literature [29]. As the data size increases, HMs regain the advantage as the quality of the $\pi_b$ estimate improves.

**Is the environment stochastic or deterministic?** While stochasticity affects all methods by straining the data requirement, HM are more negatively impacted than DM (Figure 4 center, Figure 18). This can be justified by e.g., the variance analysis of DR, which shows that the variance of the value function with respect to stochastic transitions will be amplified by cumulative importance weights and then contribute to the overall variance of the estimator; see [26, Theorem 1] for further details. We empirically observe that DM frequently outperform their DR versions in the small data case (Figure 18). In a stochastic environment and tabular setting, HM do not provide significant edge over DM, even in short horizon case. The gap closes as the data size increases (Figure 18).

### 3.3 Challenging common wisdom

To illustrate the value of a flexible benchmarking tool, in this section we further synthesize the empirical findings and stress-test several commonly held beliefs about the high-level performance of OPE methods.

**Is HM always better than DM?** No. Overall, DM are surprisingly competitive with HM. Under high-dimensionality, long horizons, estimated behavior policies, or reward/environment stochasticity, HM can underperform simple DM, sometimes significantly (e.g., see appendix Figure 18).

Concretely, HM can perform worse than DM in the following scenarios that we tested:

- Tabular with large policy mismatch, or stochastic environments (appendix Figure 18, Table 5, 8).
- Complex domains with long horizon and unknown behavior policy (app. Figure 28, 30, Table 10).

When data is sufficient, or model misspecification is severe, HM provides consistent gains over DM.

**Is horizon length the most important factor?** No. Despite conventional wisdom suggesting IPS methods are most sensitive to horizon length, we find that this is not always the case. Policy diver-

gence $\sup_{a \in A, x \in X} \frac{\pi_e(a|x)}{\pi_b(a|x)}$ can be just as, if not more, meaningful. For comparison, we designed two scenarios with identical mismatch $(\sup_{a \in A, x \in X} \frac{\pi_e(a|x)}{\pi_b(a|x)})^T$ as defined in Section 3.2 (see appendix Tables 13, 14). Starting from a baseline scenario of short horizon and small policy divergence (appendix Table 12), extending horizon length leads to $10\times$ degradation in accuracy, while a comparable increase in policy divergence causes a $100\times$ degradation.

**How good is model-based direct method (AM)?** AM can be among the worst performing direct methods (appendix Table 3). While AM performs well in tabular setting in the large data case (appendix Figure 17), it tends to perform poorly in high dimensional settings with function approximation (e.g., Figure 3 center). Fitting the transition model $P(x'|x, a)$ is often more prone to small errors than directly approximating $Q(x, a)$. Model fitting errors also compound with long horizons.

# 4 Discussion and Future Directions

Finally, we close with a brief discussion on some limitations common to recent OPE benchmarks and more generally OPE experimental studies, and point to areas of development for future studies.

*Lack of short-horizon benchmark in high-dimensional settings.* Evaluation of other complex RL tasks with short horizon is currently beyond the scope of our study, due to the lack of a natural benchmark. For contextual bandits, it has been shown that while DR is highly competitive, it is sometimes substantially outperformed by DM [56]. New benchmark tasks should have longer horizon than contextual bandits, but shorter than typical Atari games. We also currently lack natural stochastic environments in high-dimensional RL benchmarks. An example candidate for medium horizon, complex OPE domain is NLP tasks such as dialogue.

*Other OPE settings.* We outline practically relevant settings that can benefit from benchmarking:
- *Missing data coverage.* A common assumption in the analysis of OPE is a full support assumption: $\pi_e(a|x) > 0$ implies $\pi_b(a|x) > 0$, which often ensure unbiasedness of estimators [45, 33, 15]. This assumption is often not verifiable in practice. Practically, violation of this assumption requires regularization of unbiased estimators to avoid ill-conditioning [33, 16]. One avenue to investigate is to optimize the bias-variance trade-off when the full support is not applicable.
- *Confounding variables.* Existing OPE research often assumes that the behavior policy chooses actions solely based on the state. This assumption is often violated when the decisions in the historical data are made by humans instead of algorithms, who may base their decisions on variables not recorded in the data, causing confounding effects. Tackling this challenge, possibly using techniques from causal inference [50, 42], is an important future direction.
- *Strategic Environmental Behavior.* Most OPE methods have focused exclusively on single-agent scenarios under well-defined MDP. Realitic applications of offline RL may have to deal with nonstationary and partial observability induced by strategic behavior from multiple agents [62]. There is currently a lack of a compelling domain to study such a setting.

*Evaluating new OPE estimators.* For our empirical evaluation, we selected a representative set of established baseline approaches from multiple OPE method families. Currently this area of research is very active and as such, new OPE estimators have been and will continue to be proposed. We discuss several new minimax style estimators, notably the DICE family in section 2.4. A minimax-style estimator has also been recently proposed for the model-based regime [55]. Among the ideas that use marginalized state distribution [58] to improve over standard IPS, [27, 28] analyze double reinforcement learning estimator that makes use of both estimates for $Q$ function and state density ratio. While we have not included all estimators in our current benchmark, our software implementation is highly modular and can easily accommodate new estimators and environments.

*Algorithmic approach to method selection.* Using COBS, we showed how to distill a general guideline for selecting OPE methods. However, it is often not easy to judge whether some decision criteria are satisfied (e.g., quantifying model misspecification, degree of stochasticity, or appropriate data size). As more OPE methods continue to be developed, an important missing piece is a systematic technique for model selection, given a relatively high degree of variability among existing techniques.

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
