# OpenReview forum: "Empirical Study of Off-Policy Policy Evaluation for Reinforcement Learning"
_NeurIPS.cc/2021/Track/Datasets_and_Benchmarks/Round1 — NeurIPS 2021 Datasets and Benchmarks Track (Round 1)_

### Official Review · Reviewer_Eu5r · 2021-07-05
**The paper presents a benchmarking suite for the comparison of off-policy evaluation methods in several experimental settings. The suite itself is well-thought and provides a lot functionality, however, the writing of the paper could have been better.**

**Rating:** 6
**Confidence:** 3

**Strengths:**

Off-policy evaluation is a very challenging problem in the RL community and it also has significant importance for researchers in closely-related disciplines who use RL and OPE in high-risk application domains like healthcare. Therefore, the benchmarking suite presented in this paper could be of great use to the broader community.

Other than that, a strength of the paper (which is also a weakness as I discuss later on) is the breadth of the evaluated methods and the results of their comparative performance. The authors have put a lot of effort into categorizing and comparing 33 OPE methods in order to provide several insights about their relative strengths and weaknesses. I believe that the discussion of their quantitative results would be very useful for RL researchers in general.

**Weaknesses:**

As I mentioned previously, the main weakness I find in this paper is its breadth of experimental results. Even though Section 2 establishes the main characteristics of the proposed benchmarking suite, in Section 3, the train of thought starts to get lost since the authors try to evaluate and compare 33 OPE methods which are not described in detail. Therefore, it is exceptionally hard for someone not familiar with all of the original papers proposing those methods, to fully comprehend all the quantitative results presented in the paper and get some intuition about the advantages of each method. Moreover, the authors make qualitative conclusions about how different methods compare depending on the type of the experimental setting, referring to Tables and Figures in the Appendix more frequently than referring to the Tables and Figures in the main body of the paper. Overall, the experimental section makes the paper read like a review of OPE methods and it shifts the focus from the benchmarking suite itself.

The great quantity of experimental results provided by the authors make it clear that their benchmarking suite is indeed capable to create various different environments to test current OPE methods. However, I would describe the experimental section as “overwhelming”. I am not sure if it is possible, but, I would encourage the authors to focus on a few (perhaps 2-3 from each one of the discussed categories) OPE methods in the main, provide a clear description of each one of them and mainly discuss the results of those methods, having an independent discussion of the rest of the results in the Appendix (without constantly referring to the Appendix to support the claims in the main).

--- After author response ---

I read the response and I acknowledge the fact that the authors have made a decent attempt to synthesize their results and extract high-level insights from the comparison of the OPE methods. However, due to the improvements required in terms of writing (also see Clarity), I would like to keep my score (6) and encourage the authors to improve the readability of the paper for their camera-ready version if the paper gets accepted.

**Additional Feedback:**

For suggestions for improvement, look at the “Weaknesses”, “Clarity” and “Documentation” sections.

**Clarity:**

The paper could have been better in terms of writing. Specifically:
- General comment: As I said previously, having such a huge quantity of results and referring to the Appendix most of the time makes it hard for the reader to follow. The main (9 page) body should be complete on its own.
- Line 64: The authors write “The episode length T is assumed to be fixed for notational convenience”. It should be clearer, whether this is just a matter of notation or if T is assumed to be finite in the benchmarking suite. I might have missed it, but do the authors explain somewhere whether it supports infinite horizon settings or not?
- Table 1: This is a minor point but Pix-MC and Pix-GW are written as Pixel-MC and Pixel-Gw in the text.
- Line 245: The authors start to refer to Figure 4 here, however, the figure itself can be found two pages later, after Figure 3 which seems to be firstly discussed in Line 286. I think the order of the two figures could be swapped.
- Figure 3: It would be easier for the reader, if the authors added sub-captions under each panel with a short title for the content of each panel, instead of having a textual description in the caption of the figure.
- Line 286: I would encourage the authors to be consistent with the names/initials the use since the paper is already overloaded with terms. In this line, the two classes of methods are written as “DM” and “HM” but in the legend of Figure 3 they appear as “Direct” and “Hybrid”.


**Correctness:**

The evaluation methods seem to be well-thought by the authors and the experimental design is able to capture numerous OPE approaches in the literature.

**Documentation:**

The authors present sufficient detail about their implementation and the choice of hyperparameters in the appendix. Their source code is publicly available on GitHub, however, the repository itself does not seem to include a thorough guide for a user to either contribute to it or to run their own experiments. Specifically, the authors write “To run your own experiments, see example.py (or example2.py).”. I might have missed it, but, there doesn’t seem to be a description of the configuration parameters appearing in these two files.

---- After author response ---

The authors effectively addressed this comment.

**Ethics:**

Does not apply.

**Relation To Prior Work:**

The authors discuss the main advantages of their work, compared to [17,18] in Lines 51-59. The authors seem to imply that these prior pieces of work do not have the advantages of their proposed benchmarking suite. However, the paper would have benefited from a more thorough description of what [17,18] actually do, for the reader to be able to understand the importance of the current paper’s contribution.

--- After author response ---

The authors clarified this point.

**Summary And Contributions:**

The authors present COBS, a benchmarking suite for the comparison of off-policy evaluation methods in a variety of experimental settings. First, they describe their main design choices, including 6 factors which affect the performance of OPE methods and 8 environments common in the related literature which they use for their evaluation. Then, they summarize the main high-level approaches in OPE and they provide a brief overview of various already proposed methods. Lastly, they perform an extensive evaluation and comparison of those methods using their benchmarking suite and they discuss a variety of quantitative results, related to the previously discussed factors affecting the OPE performance.

---

> ### Author Response · Authors · 2021-07-13
> **Response to the review**
>
> Thank you for your comments. We appreciate your valuable feedback.
>
> First, we agree that we should expand somewhat on the related work discussion, especially [17,18]. As you rightly pointed out, off-policy evaluation is a very challenging and important sub-problem of RL, and as such, has received significant attention from the research community over the past few years. In our opinion, there has been a general lack of systematic evaluation of the relative performance of which methods work well and when. We strive to strike the right balance between (i) careful design of different settings for a comprehensive OPE benchmark and (ii) showing how one can make sense of experimental results. The attention we paid to both criteria, as well as how they may be interdependent, is the unique feature that truly sets our benchmark from standard OPE papers and prior benchmarking studies, including [17,18]. A stereotypical evaluation study may consist of selecting several domains, specifying the set of baseline methods, run the set of baselines and compare them on an aggregate basis. Note that aggregate ranking is a helpful metric (we also perform aggregation, see Fig 4 and Appendix), and it is relatively simple to present. One drawback of simple aggregation, however, is that it is easy to lose sight of the nuances that can significantly affect the performance of OPE. We spent significant parts of the paper, especially section 2 and part of section 3, to discuss exactly these nuances that motivate our benchmark design.
>
> We agree that the number of methods can seem overwhelming. This partly reflects the active nature of this research area. In choosing the set of methods for this benchmark study, we add clarity into the comparison by categorization, and by selecting the most representative among certain families of methods (otherwise, the number of methods would be even more numerous). Given quantitative comparison among OPE methods is of great interest to researchers and practitioners, some degree of “OPE review” is necessary.
>
> The point you raised that the paper shifts the focus from benchmarking to review of OPE is a valid concern for this type of study. In our paper, the results in section 3, and in particular the decision tree of factors that influences the result (fig 2) are our attempt at synthesizing an otherwise overwhelming set of quantitative results into more digestible insights. Importantly, in the context of developing the benchmark, the results in section 3 also serve the purpose of illustrating why the design choices discussed in section 2 matter.  For instance, without the mountain car scenarios, we couldn’t investigate representation mismatch as part of the decision tree in Figure 2.  We will add these discussions throughout Section 3 to connect back to the design decisions in Section 2.
> Thanks to your feedback, we will revise the paper further to better highlight the connection between our benchmark design philosophy and how to interpret the results. We will also aim to make the referenced results more self-contained within the 9 page limit.
>
> Regarding documentation, we have revised the instructions on our Github page. Previously, one of our example scripts served as a de facto tutorial for the repo. We have since added explicit instructions on how to use the repo in the main Github page. We have also added a tutorial in the form of a Jupyter notebook. We note that the benchmark is easily extensible, and there have been two separate groups of OPE researchers that have independently contributed their code to our benchmarking suite.

---

### Official Review · Reviewer_6TxH · 2021-07-05
**A benchmark with an extensive collection of OPE methods**

**Rating:** 7
**Confidence:** 4

**Strengths:**

The main strength is that the benchmark standardizes different parts of the OPE evaluation pipeline which is an intensive and largely avoidable work. For example, the interfaces among policy, environment, and OPE models; and the evaluation metrics. This will greatly reduce the time needed to setup the pipeline for future work and ease benchmarking against standard methods. The IPS, direct, and hybrid methods implemented in the benchmark are seldom compared against each other, and seldom evaluated in domains that cover different dimensions of complexity of the OPE problem. Thus, the benchmark is a useful contribution for finding methods that work well and identifying their limitations.

**Weaknesses:**

The main weakness is that there is little supporting documentation and tools to make it easier to use the benchmark for new OPE methods. The repository [https://github.com/clvoloshin/COBS] seems to provide three example scripts to get started, which are not documented. It is left to the reader to figure out the interfaces and extend them for their own models, domains, and evaluation metrics. This increases the effort required to use the benchmark and decreases the chances of its wide adoption. A well documented API and an example tutorial [e.g. the one in bsuite https://github.com/deepmind/bsuite] will greatly enhance this benchmark’s utility to the community. Further, the extensibility of the benchmark to include new OPE methods will be important given the rapid advances being made on the problem.

**Additional Feedback:**

As an earlier implementation of the benchmark [Voloshin et al. 2019 https://arxiv.org/abs/1911.06854] has been available for some time now, can authors describe the ways in which it has been used in the past? What are the changes made to the earlier version and how they will increase its impact on the community?

Based on my experience using the GitHub code, I believe, that the benchmark has useful features. But documenting the features and adding tutorials for getting started will greatly benefit its usability. I am open to increase my ratings if authors provide evidence of the ease of usage and extensibility of the benchmark.

Minor questions on extensibility of the benchmark:

Is it easy to extend the benchmark to include contextual bandit domains and leverage existing implementations of OPE methods?

Are there provisions to add new evaluation metrics? For example, risk measures (e.g. CVaR or other summaries of the cumulative return used in https://arxiv.org/abs/2104.12820) to test robustness of the policies.

Going forward, are there plans to synthesize results from third-parties using the benchmark to evaluate new methods?

**Clarity:**

The paper is written clearly. The design choices are well motivated. Experimental results are synthesized very succinctly in a decision tree to provide guidelines for best methods for different problem types.

**Correctness:**

Evaluation methodology is well documented and sound. Different factors that influence OPE methods’ performance such as horizon length, dimensionality of state space are varied across the included domains.

**Documentation:**

The implemented methods are described in enough detail. Code for generating the datasets, and learning and evaluating models are publicly accessible. Scripts for reproducing the reported results are provided. Documentation of the scripts and different components of the benchmark needs major improvement.

**Ethics:**

No ethics implications are highlighted by the authors (except energy use from extensive computations). I concur that there are no ethics implications but would suggest adding a disclaimer about the scope of claims that can be made using this benchmark. This is to avoid exaggerated claims of safety given that OPE methods are critical for evaluating algorithms for high-risk scenarios like medical treatments.

**Relation To Prior Work:**

Improvements over prior benchmarks are highlighted. The benchmark provides a diverse set of domains and enables simulating evaluation under a variety of settings.

**Summary And Contributions:**

The work introduces a benchmark for off-policy policy evaluation (OPE) methods which is the problem of predicting future performance of a policy without actually deploying the policy and given only historical data. The benchmark contributes eight reinforcement learning domains of varying complexity and implementations of a variety of standard evaluation methods.

===
After the response

Thank you adding the example tutorial and more documentation. I would encourage authors to keep working on improving ease of use - by adding more end-to-end, stand-alone tutorials (and not just pointing to code sections to change), by moving to an automatic documentation generator from docstrings, and explaining the example scripts. Further, host some of these on a website, including the benchmarked results from the paper for each dataset setting.

---

> ### Author Response · Authors · 2021-07-13
> **Responses to the review**
>
> Thank you for using parts of the code! We also thank you for pointing out that more documentation is necessary and, in response, we have substantially updated the README as well as created a commented Jupyter Notebook tutorial Tutorial.ipynb. Previously the script example.py served as the de-facto tutorial for the benchmark. Now we have made the tutorial more clear and explicit.
>
> Voloshin et al. 2019 is an unpublished work that has an ablation study focus. We believe that the current NeurIPS Benchmarks & Datasets Track is the appropriate venue to highlight the suite component.
>
> Our codebase is easily extensible. Contextual bandit domains can be incorporated so long as the domains are wrapped to take the open-ai gym interface. Depending on the complexity of the performance metric, it is fairly easy to swap in new metrics. We will document this as well.
>
> The goal is to provide researchers the tools and reference implementations of these methods and we hope that third-parties will contribute back to the suite with their own reference implementations as well. We note that so far, at least two groups of RL researchers have actively used our suite to benchmark their own algorithms. One group in particular from Cornell has further contributed their own method to COBS. This demonstrates the benchmarking suite’s usefulness and ease of use. We intend to continue the improvement of documentation as more environments and methods are added in the future. Thank you for your feedback.

---

### Official Review · Reviewer_74Rw · 2021-07-06
**A reasonable benchmark for Off-Policy Evaluation**

**Rating:** 7
**Confidence:** 3
**Correctness:** Yes, the dataset and evaluation seem …

**Strengths:**

The paper presents a benchmark dataset which can be used by the community. It also presents evaluations that may be useful for off policy methods in practice.

**Weaknesses:**

I think the benchmark dataset could be broadened to include tasks that are more similar to how off-policy evaluation may be used in practical settings.

**Additional Feedback:**

N/A

**Clarity:**

I think the paper could be edited for better readability particularly in the results sections- with so many methods and abbreviations it is a bit difficult to parse.

**Documentation:**

I didn't see any hosting licensing and maintenance plan for the benchmark.

**Ethics:**

No.

**Relation To Prior Work:**

Yes.

**Summary And Contributions:**

This paper describes the release of a dataset and a benchmarking of off-Policy reinforcement learning methods. A comment I would have about the proposed dataset is that I wish that the dataset had tried to include some more tasks that are similar to ones in which off-policy evaluation would be used in the real world- either a more complex simulated robotics task perhaps, or perhaps from another domain entirely- online eduction, search recommendation systems, etc.
In Table 1, I'm not sure why Q class would be an aspect of the environment necessarily, as any environment could be attempted to be modeled using any Q class.
I think the results section would have been more readable has there been a table of all methods evaluated with the abbreviations used to denote that method. The large number of abbreviations used are difficlt to cross-reference in the methods section. In Figure 4, there should be a description of how "Near-top Freq." is calculated, exactly.
The discussion section is clear and well-written, and provides clear general guidelines for selection amongst different off-policy methods.

---

> ### Author Response · Authors · 2021-07-13
> **Response to the review**
>
> Thank you for your comments. We selected the environments carefully in order to allow researchers a systematic evaluation of the relative performance of which methods work well and when. We agree adding more realistic environments is useful and interesting for future research and we look forward to collaborating with those who have access to such domains.
>
> You are correct to point out that the Q function is not really part of the environment, but we wanted to show which Q functions we used in which environment.  We can make this point more clear in the writing.
>
> Space permitting, we will add a table of all methods and abbreviations to add clarity. The appendix contains quite a bit more information, including detailed explanations of all methods used.
>
> Near-top Freq is one of the metrics discussed in Section 2.3.
>
> We appreciate your feedback to improve the readability of the paper and will reflect them in the revision.

---

### Decision · Program_Chairs · 2021-07-26

**Decision:**

Accept

**Comment:**

All reviewers vote for accepting the paper. One concern in multiple reviews was the clarity of the writing. In addition, one reviewer also pointed out shortcomings in the documentation (which the authors have addressed by now). Hence I encourage the authors to carefully edit the paper for clarity. Overall I recommend accepting the paper since it fills a need for a common benchmark in off-policy policy evaluation.